

# Differentiation between maxillary and malar midface position within the facial profile

Chimène Chalala[1],* and Joseph G. Ghafari[2],*

[1] Departments of Orthodontics and Dentofacial Orthopedics, Lebanese University and American University of Beirut, Beirut, Lebanon
[2] Division of Orthodontics and Dentofacial Orthopedics, American University of Beirut, Beirut, Lebanon
* These authors contributed equally to this work.

## ABSTRACT

**Aims:** To define midfacial position differentiating maxillary and zygomatic regions and to evaluate the corresponding cephalometric characteristics discerning midfacial flatness and fullness.

**Material and Methods:** A total of 183 pretreatment lateral cephalometric radiographs of non-growing orthodontic patients (age 25.98 ± 8.43 years) screened at our university orthodontic clinic. The lateral cephalographs of the orthodontic patients were stratified in four groups: *flat, normal toward flat, normal toward full, full,* according to distances from nasion and sella to points J and G (NJ, SJ, NG and SG). J is the midpoint of the distance connecting orbitale to point A, and G the center of the triangle connecting orbit, key ridge and pterygomaxillary fissure. Statistics included the Kendall tau-b test for best associations among measurements.

**Results:** All measurements were statistically significantly different between *flat* and *full* groups. The highest associations were between NJ and SJ ($\tau b$ = 0.71; $p < 0.001$) and NG and SG ($\tau b$ = 0.70; $p < 0.001$). Flat midfaces were characterized by canting of the cranial base and palatal plane, hyperdivergent pattern and maxillary retrognathism. The opposite was true for fuller midfaces.

**Conclusion:** Midface skeletal location was assessed differentially in the naso-maxillary and malo-zygomatic structures differentially. Craniofacial characteristics were identified according to this stratification, indicating the potential for application in facial diagnosis and need for testing on 3D cone-beam computed tomography images.

Corresponding author
Chimène Chalala,
chalalachimene@yahoo.com

## INTRODUCTION

The location of the midface within the facial profile has been mostly assessed in relation to the maxilla, thus linking maxillary prognathism or retrognathism to associated malocclusions. The position of the maxilla has served as the only practical cephalometric measurement for midface flatness or fullness through the relative position of point A to the cranial base (SNA angle or position of A to nasion perpendicular) (*McNeill, Proffit & White, 1972*; *Steiner, 1953*; *Downs, 1949*; *McNamara, 1984*; *Jarabak & Fizzel, 1972*;

*Ricketts, 1960*; *Ricketts, 1961*). Given the restrictive two-dimensional nature of cephalometric analysis, researchers and clinicians focused on this midline landmark (point A) without consideration of the position of the malar bone (cheekbone), which is another essential determinant of midfacial flatness or fullness. Indeed, the midface encompasses the region between the zygoma and maxilla horizontally, and vertically between the eyebrows and subnasale horizontal planes (*Zide, Grayson & McCarthy, 1981*).

In planning orthognathic surgery of maxillary retrognathism associated with Class III malocclusion, 2D and 3D simulation programs are based on the movement of the maxilla, with the knowledge that maxillary advancement through the Lefort 1 osteotomy would also lead to fuller cheekbone appearance (*Petersen, Markiewicz & Miloro, 2018*). In plastic surgery, midface cheek hollowness has been corrected with the placement of submalar implants addressing the flatness at the zygomatic area (*Kridel & Patel, 2017*).

Most of the research conducted to examine the presence and treatment of dentoskeletal deformities had focused primarily on the lower facial region (*Brooks et al., 2001*; *Kolokitha & Topouzelis, 2011*; *Betts et al., 1993*), with scarce analyses of midface position as an entity regardless of the type of malocclusion. *Singh, McNamara & Lozanoff (1998)* evaluated morphometrically the midfacial deficiency in Class III compared with Class I patients through seven landmarks in the maxilla, but none related to the malar bone. *Zide, Grayson & McCarthy (1981)* added orbitale, off the midline, to the regular midline landmarks nasion and A point, to assess midfacial deficiency. Using cone-beam computed tomography (CBCT) derived multiplanar-reconstructed cross-sections, *Kim et al. (2018)* defined differences between male and female soft tissue midface deficiency in Class III malocclusion. We hypothesized that hard tissue differentiation is possible between maxillary and malar positions by identifying representative geometric landmarks within each of these structures corresponding to the different planes in which these structures are located. Key to such identification was to define both the maxillary and malar planes lateral to the midsagittal plane. Given the limitation by Institutional Review Boards in using CBCT imaging as a routine record, we initiated this evaluation on regular 2-D cephlograms as a first step to test this hypothesis.

The aims of this study were to evaluate: (1) the differential anteroposterior position of the maxillary and zygomatic regions, and the correspondence in diagnosis between these regions, and (2) the corresponding cephalometric characteristics and measurements discerning midfacial flatness and fullness.

## MATERIALS AND METHODS

Prior to data collection, the study was approved by (our) university's Institutional Review Board (IRB ID: BIO-2018-0065). Selected from the database of orthodontic records at (our) Medical Center, the pretreatment lateral cephalometric radiographs of 183 non-growing healthy patients (mean age 25.98 ± 8.43 years; 106 females, 77 males) belonged to patients whose treatment has been initiated or completed. The initial selection criterion was skeletal age, which was determined through the cervical vertebrae maturation (CVM) assessment method as modified by *Baccetti, Franchi & McNamara (2005)*.

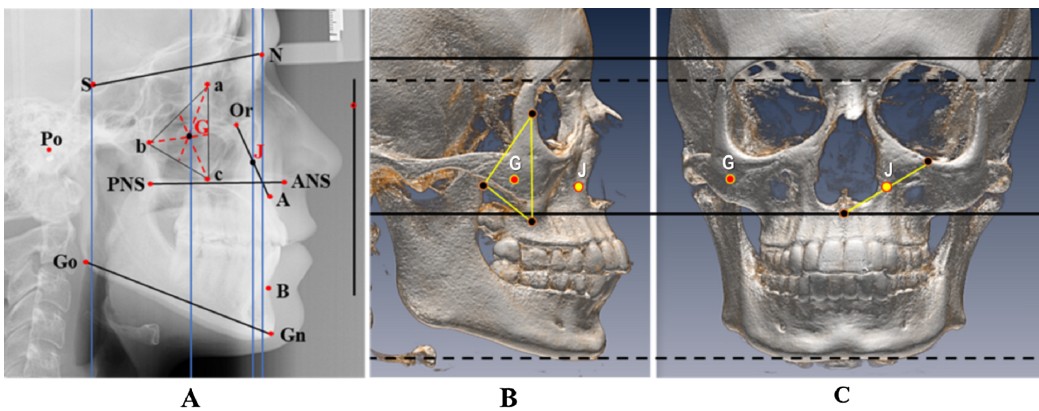

**Figure 1  Cephalometric landmarks and measurements, with two new landmarks defining midfacial flatness.** (A) Cephalometric landmarks and measurements, with two new landmarks defining midfacial flatness: J, the midpoint of the distance connecting orbitale to point A, and G, the center of maxillary triangle: a: the dorsal surface of the orbit in the infratemporal fossa, b: the deepest point on the curved anterior surface of the pterygomaxillary fissure, c: the lowest point on the outline of the zygomatic process. Vertical: corresponds to natural head position. Landmarks: N, nasion; S, sella; Go, gonion; Gn, gnathion; B, B point; A, A point; ANS, anterior nasal spine; PNS, posterior nasal spine; Or, orbitale; Po, porion; J: midpoint of the distance connecting orbitale point to A point; G: center of maxillary triangle; a: posterior border of the orbit; b: deepest point of the anterior curvature of pterigomaxillary fissure; c: lowest point of the key ridge. Measurements: SN: anterior cranial base; PP: palatal plane; ANS–PNS; MP: mandibular plane; Go-Gn. Angles: SN to Horizontal (H); PP to Horizontal (H); MP to Horizontal (H); SNA; SNB; ANB; SN-MP; Maxillary depth: angle between Frankfort plane and NA line. Linear flatness measurements : S-J; N-J; S-G; N-G, all taken between the projections of J and G on the vertical lines through nasion and sella. (B and C) Graphs illustrating on sagittal (B) and frontal (C) views the virtual geometric positions of: a. point J (in yellow), nearly in the center of the naso-maxillary complex in the transverse plane and midway between orbit and palatal plane in the vertical dimension, essentially capturing the position of the maxilla in the midface, and b. point G (in red), nearly in the center of the malo-zygomatic complex between the posterior wall of the orbit, the pteyrygomaxillary plate and the key ridge, essentially defined by the lateral boundaries of the maxilla and representing the cheekbone projection in the midface. The horizontal lines indicate the correspondence of landmarks between the lateral and frontal skull views.               

Accordingly, the minimum age for inclusion was 16.2 years for females, and 18.1 years for males.

Subjects were excluded from the study if they had a cervical stage <6 (because the highest CVM score of six indicates cessation of mandibular growth for at least 2 years), previous orthodontic treatment, any craniofacial anomaly (patients with syndromes such as cleft lip/palate, hemifacial macrosomia, Treachers Collins, plagiocephaly and other congential malformations), or if their radiographs were of non-diagnostic quality.

Lateral cephalometric radiographs were taken with the head oriented in natural head position following a standard procedure in the same cephalostat (GE, Instrumentarium, Tuusula, Finland). Selected landmarks were digitized (Fig. 1) and variables measured by one operator (CC) using the View box 4 software (dHAL Software, Kifissia, Greece). The radiographs were calibrated in reference to the ruler present on the lateral cephalograms.

**Table 1 Distribution of four groups according to the position of J and G relative to nasion and sella.**

| Measurement Mean-SD | Flat | Normal toward flat | Normal toward full | Full |
|---|---|---|---|---|
| SJ/SN × 100 91.46 ± 4.81 | <86.65 (n = 35) | 86.65–91.46 (n = 52) | 91.46–96.27 (n = 69) | >96.27 (n = 27) |
| SG/SN × 100 53.45 ± 3.25 | <50.2 (n = 29) | 50.2–53.45 (n = 64) | 53.45–56.7 (n = 62) | >56.7 (n = 28) |
| NJ/SN × 100 6.41 ± 5.68 | >12.09 (n = 29) | 6.41–12.09 (n = 60) | 0.73–6.41 (n = 70) | <0.73 (n = 24) |
| NG/SN × 100 44.41 ± 4.18 | >48.59 (n = 26) | 44.41–48.59 (n = 60) | 40.23–44.41 (n = 74) | <40.23 (n = 23) |

To define the separate locations of the maxilla-nasal and malo-zygomatic regions, two geometric landmarks were determined, akin to the geometric definition of point S for the center of sella turcica (Fig. 1):

a. J, the midpoint of the distance connecting points orbitale (O) and A. The rationale for this selection is that flatness or fullness is associated with upper (orbitale) and lower (point A) components of the maxillary structure, J representing the central area of the medial part of the maxilla.

b. G, the geometric center of the maxillary triangle described in the Moorrees mesh diagram analysis (Ghafari, 2006) and which connects the posterior border of the orbit with the inferior outline of the key ridge (zygomaxillare) and the anterior surface of the pterygomaxillary fissure, which lines up with the malo-zygomatic suture (Fig. 1). By this definition, G is a more lateral landmark that would represent the central malar area.

Consequently, J and G approximated the positions of the mid-maxillary and malar bones, respectively.

Linear measurements were taken of the projections of J and G on vertical lines drawn anteriorly through nasion and posteriorly through sella. The measurements were SJ, SG, NJ and NG (Fig. 1). Given differences in head size among individuals and between genders, these distances were scaled to the cranial base length (SN), and the resulting ratios (SJ/SN × 100, SG/SN × 100, NJ/SN × 100, NG/SN × 100) were calculated. Greater distances between either J or G to the anterior landmark nasion would indicate a tendency toward midface flatness and shorter distances would indicate a direction toward midfacial fullness. The opposite would apply for the distances between either J or G and the posterior landmark sella. Accordingly, four groups were generated based on the mean and standard deviations (SDs) of each calculated ratio (Table 1):

1. *Flat group (f)* included the subjects in whom the ratio was less than one SD for the measurements made relative to the posterior landmark sella and beyond one SD for the measurements calculated relative to the anterior landmark nasion.

2. *Normal toward flat group (Nf)* included the subjects in whom the ratio was between the mean and −1 SD for J and G measurements taken relative to sella, and between the mean and + 1 SD for measurements calculated relative to nasion.

3. *Normal toward full group (NF)* included the subjects whose ratio was between the mean + 1 SD for J and G measurements computed to sella, and between the mean + 1 SD for the measurements calculated relative to nasion.

4. *Full group (F)* included the subjects whose ratio was beyond +1 SD for the measurements calculated to sella and <1 SD for the measurements computed relative to nasion.

Cephalometric measurements of the relationships among cranial base and jaws in the sagittal and vertical planes were compared among these groups. Measurements on all the variables were repeated at different times by the first author on 30 radiographs to assess intra-examiner reliability, and by the second author for inter-rater reproducibility.

## Statistical analysis

The repeated measures were evaluated with the two-way mixed effects intra-class correlations for absolute agreement on single measures. A one-way between subjects analysis of variance (ANOVA) was applied to study differences of selected cephalometric measurements among the four groups. A Kendall's tau-b correlation was employed to determine the relationship between measurements among these groups. The Pearson product-moment correlation was calculated to determine the association among variables, and the intra-class coefficients were calculated for intra and inter examiner reliability.

As part of post hoc power assessment, effect sizes were calculated for all cephalometric variables across the four groups. For all variables, effect size ranged from 0.543 for ANS–PNS in the NG comparisons to 2.84 for SN/H for the same comparisons (large effect size). Using the total sample size of 183, the calculated post hoc power ranged between 0.999 and 1.000 for all ANOVA comparisons. However, since this power calculation assumes equal sample sizes, a conservative approach was used to estimate power by using the smallest group sample size and multiplying by 4, that is, $n = 96$ for NJ and $n = 92$ for NG. Using this conservative approach, post hoc sample size varied between 0.995 and 1.000 for all ANOVA comparisons. Accordingly, the sample size was amply sufficient to perform the group comparisons.

The IBM® SPSS® 23.0 statistical package was used to carry out all statistical analyses. Statistical significance was set at 0.05.

## RESULTS

Intra-class correlation coefficients ranged between $0.963 < r < 0.996$ for intraexaminer measurements, and $0.924 < r < 0.962$ for inter-examiner measurements, indicating high correspondence of the repeated measures and accurate reproducibility of landmark identification. The results are presented in four sets of classification based on the four defined groups. Under all classifications, the group comparisons revealed statistically significant differences for nearly all the measurements between *flat* and *full* groups. The Kendall tau-b test for best associations among the measurements classified on the distances NJ, SJ, NG and SG over SN (Table 2) revealed the highest associations between measurements assessed by NJ/SN and SJ/SN ($\tau b = 0.71$; $p < 0.001$) and those by NG/SN

**Table 2 Kendall's tau-b test.** Kendall's tau-b among cephalometric measurements* and concordance percentage (%) among defined categories.

|  | NJ/SN | NG/SN | SJ/SN |
|---|---|---|---|
| NJ/SN |  |  |  |
| NG/SN | 0.687 (88%) |  |  |
| SJ/SN | 0.712 (87%) | 0.482 (78%) |  |
| SG/SN | 0.481 (78%) | 0.701 (87%) | 0.53 (79%) |

**Note:**
* Based on group classifications (f, Nf, NF, F) according to the distances from J and G to nasion (N) and sella (S).

and SG/SN measurements ($\tau b$ = 0.70; $p$ < 0.001). The associations were lower for SJ/SN and NG/SN measurements ($\tau b$ = 0.48; $p$ < 0.001); NJ/SN and for SG/SN measurements ($\tau b$ = 0.48; $p$ < 0.001). The concordance (%) of categorization among groups was not total (100%) yet high (79–88%). The discordance ranged between 8% and 22%, highest between the normal groups (Nf and NF) and nil (0%) between the extreme groups F and f. However, when the normal groups were grouped into one group (N), only 5% to 11% discordance was observed between f and N, and between F and N.

Accordingly, the outcomes defined by NJ/SN and NG/SN are displayed for group comparisons (Tables 3 and 4). The main findings from these comparisons are presented in relation to the cephalometric structures and relations:

1. *Cranial base*: the cant of the cranial base (SN/H) was statistically significantly greater (lower sella) in the *full* group relative to the *flat* group for both the NJ/SN set (14.70 ± 4.61; 5.64 ± 3.39 respectively, Table 3) and the NG/SN set (15.74 ± 3.85; 5.40 ± 3.11 respectively, Table 4).

2. *Jaw positions and relations*: The SNA angle was smaller in the *flat* group compared with the *full* group in both the NJ/SN set (79. 67 ± 3.85; 85.30 ± 3.79, respectively, Table 3), and in the NG/SN set (80.06 ± 3.76; 83.86 ± 4.34, respectively, Table 4). No statistically significant difference was observed in the SNB angle among all groups. The ANB angle was statistically significantly smaller in the *flat* group compared with the *full* group in both the NJ/SN set (−2.462 ± 3.5170; 4.854 ± 3.5741 respectively, Table 3) and the NG/SN set (−1.519 ± 3.5846; 4.435 ± 3.7640 respectively, Table 4).

3. *Jaw-specific measurements*: The cant of the palatal plane (PP/H) was statistically significantly tipped down posteriorly with a fuller midface compared to a flat midface in the NJ/SN set (5.85 ± 4.15; −2.57 ± 3.87 respectively, Table 3), and the NG/SN set (6.50 ± 3.60; −3.26 ± 3.22, respectively, Table 4). As gauged by the maxillary depth (NA/FH) angle, less maxillary depth was observed in flat midfaces. The angle was statistically significantly lower in the *flat* group compared with the *full* group in the NJ/SN set (88.58 ± 3.49; 93.68 ± 3.73 respectively, Table 3), and the NG/SN set (88.91 ± 2.87; 93.17 ± 3.99 respectively, Table 4).

The mandibular plane was steeper in the *flat* group, with the mandibular plane angle statistically significantly decreased in the *full* group relative to the *flat* group in the NJ/SN

**Table 3 Comparison of measurements among groups as defined by NJ/SN × 100 (n = 183).**

**(A) Analysis of Variance (ANOVA test)**

| NJ/SN × 100 | Flat >12.09 | | Normal to Flat 6.41–12.09 | | Normal to Full 0.73–6.41 | | Full <0.73 | | ANOVA | |
|---|---|---|---|---|---|---|---|---|---|---|
| n | 29 | | 60 | | 70 | | 24 | | | |
| | Mean | SD | Mean | SD | Mean | SD | Mean | SD | F | p |
| Sagittal measurements | | | | | | | | | | |
| SN/H | 5.64 | 3.39 | 10.59 | 3.90 | 11.88 | 3.85 | 14.70 | 4.605 | 26.64 | <0.001** |
| SNA | 79.67 | 3.85 | 80.51 | 4.54 | 83.30 | 3.44 | 85.30 | 3.794 | 14.39 | <0.001** |
| SNB | 82.10 | 4.59 | 78.88 | 5.01 | 79.30 | 4.09 | 80.44 | 5.173 | 3.59 | 0.015* |
| ANB | −2.46 | 3.52 | 1.62 | 3.96 | 4.00 | 3.29 | 4.85 | 3.574 | 26.95 | <0.001** |
| Maxillary depth | 88.58 | 3.49 | 89.86 | 3.59 | 92.52 | 3.02 | 93.68 | 3.728 | 16.78 | <0.001** |
| ANS–PNS | 54.31 | 3.69 | 52.41 | 4.00 | 53.25 | 4.25 | 53.56 | 4.369 | 1.51 | 0.214 |
| Vertical measurements | | | | | | | | | | |
| PP/H | −2.57 | 3.87 | 1.72 | 4.11 | 2.87 | 3.69 | 5.85 | 4.15 | 22.13 | <0.001** |
| MP/H | 22.34 | 5.72 | 19.72 | 6.32 | 19.88 | 5.73 | 14.60 | 5.136 | 8.01 | <0.001** |
| PP/MP | 19.79 | 7.01 | 21.45 | 7.01 | 22.754 | 6.57 | 20.44 | 6.40 | 1.62 | 0.187 |
| SN/MP | 28.01 | 7.13 | 30.31 | 7.52 | 31.753 | 6.63 | 29.29 | 6.59 | 2.20 | 0.090 |
| Upper AFH | 52.84 | 2.73 | 51.82 | 3.14 | 51.30 | 3.37 | 50.42 | 2.95 | 2.92 | 0.036* |
| Lower AFH | 66.58 | 7.57 | 64.42 | 7.40 | 64.65 | 6.85 | 63.76 | 5.91 | 0.859 | 0.464 |
| Total AFH | 119.42 | 8.82 | 116.22 | 8.43 | 115.95 | 8.26 | 114.18 | 6.65 | 1.96 | 0.12 |
| UAFH/TAFH | 44.38 | 2.70 | 44.70 | 2.96 | 44.34 | 2.76 | 44.23 | 2.70 | 0.25 | 0.86 |
| LAFH/TAFH | 55.62 | 2.70 | 55.30 | 2.96 | 55.65 | 2.76 | 55.77 | 2.70 | 0.25 | 0.86 |
| UAFH/LAFH | 80.23 | 8.88 | 81.37 | 9.85 | 80.09 | 8.96 | 79.72 | 8.68 | 0.29 | 0.83 |

**(B) Post hoc comparisons of statistically significant measurements by ANOVA**

| | ANOVA p | Flat vs. Normal flat | Flat vs. Normal full | Flat vs. Full | Normal flat vs. Normal full | Normal flat vs. Full | Normal full vs. Full |
|---|---|---|---|---|---|---|---|
| SN/H | <0.001** | <0.001** | <0.001** | <0.001** | 0.379 | <0.001** | 0.016* |
| SNA | <0.001** | 1.000 | <0.001** | <0.001** | 0.001** | <0.001** | 0.192 |
| SNB | 0.015* | 0.015* | 0.040* | 1.000 | 1.000 | 1.000 | 1.000 |
| ANB | <0.001** | <0.001** | <0.001** | <0.001** | 0.001** | 0.002** | 1.000 |
| Maxillary depth | <0.001** | 0.590 | <0.001** | <0.001** | <0.001** | <0.001 | 0.904 |
| PP/H | <0.001** | <0.001** | <0.001** | <0.001** | 0.574 | <0.001** | 0.009** |
| MP/H | <0.001** | 0.298 | 0.355 | <0.001** | 1.000 | 0.002** | 0.001** |
| UAFH | 0.036* | 0.910 | 0.17 | 0.036* | 1.000 | 0.41 | 1.000 |

Notes:
H, horizontal; maxillary depth, angle between Frankfort horizontal and NA; AFH, anterior face height.
* Statistically significant at $p < 0.05$.
** Statistically significant at $p < 0.01$.

set (14.60 ± 5.14; 22.34 ± 5.72 respectively, Table 3), and the NG/SN set (14.70 ± 5.92; 22.23 ± 4.77 respectively, Table 4). The upper anterior face height (AFH) was statistically significantly greater in the *flat* groups (Tables 3 and 4).

**Table 4 Comparison of measurements among groups as defined by NG/SN × 100 (n = 183).**

**(A) Analysis of variance (ANOVA test)**

| NG/SN × 100 | Flat >48.59 | | Normal to Flat 44.41–48.59 | | Normal to Full 40.23–44.41 | | Full <40.23 | | ANOVA | |
|---|---|---|---|---|---|---|---|---|---|---|
| n | 26 | | 60 | | 74 | | 23 | | | |
| | Mean | SD | Mean | SD | Mean | SD | Mean | SD | F | p |
| Sagittal measurements | | | | | | | | | | |
| SN/H | 5.396 | 3.112 | 9.872 | 3.769 | 12.012 | 3.82 | 15.735 | 3.854 | 35.715 | <0.001** |
| SNA | 80.058 | 3.759 | 81.687 | 4.674 | 82.523 | 4.03 | 83.861 | 4.34 | 3.734 | 0.012* |
| SNB | 81.550 | 4.821 | 79.187 | 5.046 | 79.692 | 4.06 | 79.417 | 5.562 | 1.595 | 0.192 |
| ANB | −1.519 | 3.585 | 2.500 | 4.471 | 2.835 | 3.77 | 4.435 | 3.764 | 10.633 | <0.001** |
| Maxillary depth | 88.912 | 2.873 | 90.700 | 3.682 | 91.738 | 3.723 | 93.165 | 3.99 | 6.583 | <0.001** |
| ANS–PNS | 53.992 | 2.744 | 53.635 | 4.344 | 52.711 | 4.182 | 52.587 | 4.54 | 1.062 | 0.367 |
| Vertical measurements | | | | | | | | | | |
| PP/H | −3.262 | 3.216 | 1.417 | 4.331 | 2.974 | 3.399 | 6.500 | 3.599 | 30.535 | <0.001** |
| MP/H | 22.227 | 4.768 | 20.372 | 5.618 | 19.395 | 6.409 | 14.700 | 5.924 | 7.404 | <0.001** |
| PP/MP | 18.988 | 5.622 | 21.803 | 6.814 | 22.361 | 6.9 | 21.191 | 7.326 | 1.648 | 0.180 |
| SN/MP | 27.638 | 5.54 | 30.248 | 7.021 | 31.392 | 7.241 | 30.435 | 7.798 | 1.837 | 0.142 |
| Upper AFH | 52.75 | 2.53 | 51.96 | 3.13 | 51.37 | 3.35 | 50.07 | 3.04 | 3.378 | 0.020* |
| Lower AFH | 64.01 | 6.68 | 64.56 | 7.29 | 65.50 | 7.20 | 63.76 | 6.31 | 0.537 | 0.658 |
| Total AFH | 116.75 | 7.89 | 116.52 | 7.91 | 116.86 | 9.01 | 113.83 | 7.26 | 0.828 | 0.480 |
| UAFH/TAFH | 45.29 | 2.47 | 44.72 | 3.14 | 44.06 | 2.55 | 44.07 | 2.76 | 1.617 | 0.187 |
| LAFH/TAFH | 54.71 | 2.47 | 55.28 | 3.15 | 55.94 | 2.55 | 55.91 | 2.77 | 1.592 | 0.193 |
| UAFH/LAFH | 83.15 | 8.45 | 81.49 | 10.42 | 79.12 | 8.21 | 79.23 | 8.89 | 1.678 | 0.173 |

**(B) Post hoc comparisons of statistically significant measurements by ANOVA**

| | ANOVA p | Flat vs. Normal flat | Flat vs. Normal full | Flat vs. Full | Normal flat vs. Normal full | Normal flat vs. Full | Normal full vs. Full |
|---|---|---|---|---|---|---|---|
| SN/H | <0.001** | <0.001** | <0.001** | <0.001** | 0.007** | <0.001** | <0.001** |
| SNA | 0.012* | 0.630 | 0.072 | 0.013* | 1.000 | 0.232 | 1.000 |
| ANB | <0.001** | <0.001** | <0.001** | <0.001** | 1.000 | 0.297 | 0.569 |
| Maxillary depth | <0.001** | 0.226 | 0.005** | <0.001** | 0.614 | 0.038 | 0.612 |
| PP/H | <0.001** | <0.001** | <0.001** | <0.001** | 0.104 | <0.001** | 0.001** |
| MP/H | <0.001** | 1.000 | 0.217 | <0.001** | 1.000 | 0.001** | 0.006** |
| UAFH | 0.020* | 1.000 | 0.333 | 0.020 | 1.000 | 0.090 | 0.512 |

Notes:
H, horizontal; maxillary depth, angle between Frankfort horizontal and NA; AFH, anterior face height.
* Statistically significant at $p < 0.05$.
** Statistically significant at $p < 0.01$.

In general, a progression was noted among the statistically significant measurements ascending from the *flat* group toward the *full* group.

For the total ungrouped sample, the Pearson correlation coefficients were high between NJ/SN and NG/SN ($r = 0.88$, $p < 0.001$), and between SJ/SN and SG/SN ($r = 0.83$, $p < 0.001$).

## DISCUSSION

The findings revealed the potential for differentiating flatness and fullness of the maxillary and the zygomatic areas through distinctive linear measurements made from landmarks defined within these areas to anterior (nasion) and posterior (sella) cranial base landmarks. This delineation provides a quantitative basis for the assessment of the midface at two levels lateral to the midline, complementing the prevalent method used in practice and prior studies in which the anteroposterior location of the midface was mostly related to the position of the maxilla (*McNeill, Proffit & White, 1972*; *Steiner, 1953*; *Downs, 1949*; *McNamara, 1984*; *Jarabak & Fizzel, 1972*; *Ricketts, 1960*; *Ricketts, 1961*), often specifically associated with the midsagittal SNA angle. The assessment also supplements methods employed in anthropology, whereby flatness or fullness of the midface has been defined in the frontal plane, mainly by the projection of point A between zygomaxillary anterius right and left points (*Yamaguchi, 1973*; *Yamaguchi, 1980*).

The measurements used to define the groups were different from those used to assess other pertinent characteristic measures. Regardless of the method used for stratifying the four groups in gradation from flatness through fullness, the cephalometric measurements revealed the following:

1. *flat* groups were characterized by a higher position of sella and posteriorly tipped-up palatal plane, features found in Class III malocclusion, which in turn has been associated with maxillary retrognathism and presumed associated midface flatness (*Ghafari, Haddad & Saadeh, 2011*). In addition, the midface position was not related to the length of the maxilla as ANS–PNS was similar across all groups. This finding suggests that the position of the maxilla rather than its midsagittal length is associated with midface flatness or fullness. In this context, the subspinale region was in a more backward position in *flat* groups with less maxillary depth.

2. Steepness of the mandibular plane was more significant in *flat* groups, portraying a hyperdivergent pattern. The increased mandibular plane angle has been previously reported as the most reliable indicator in assessing facial vertical growth pattern (*Ahmed, Shaikh & Fida, 2016*). The finding that the upper AFH was greater in the flat groups may also be related to the association with mandibular hyperdivergence and steepness of the palatal plane.

3. The SNB angle was not statistically significantly different among the groups, indicating that the methods of assessing midface position were typically related to the maxilla and did not discriminate the position of the mandible.

Several findings indicate the suitability of the used measurements to properly reflect midface position:

1. Definite characteristics often clinically associated with either midface flatness or fullness were actually determined with the measurements to J and G, such as the ANB angle being smaller in the *flat* group, as would be expected in a Class III malocclusion, and the opposite holding for increased ANB in the *full* group, more commensurate with Class II

malocclusion. To gauge the applicability of the findings, the methods used in this study were applied in a representative number of patients with variations of midface characteristics. The cephalographs and facial profiles of some of these patients were superimposed and are shown in Fig. 2. Research targeting different malocclusion groups is warranted to further confirm this conclusion.

2. All the statistically different measurements disclosed a progression from *flat* toward *normal* and *full* groups, actually differentiating flat and full midfaces.

3. The correspondence between conclusions based on anterior (NJ/SN and NG/SN) and posterior (SJ/SN and SG/SN) measurements. Yet, given the higher correlations between NJ/SN and SJ/SN, and between NG/SN and SG/SN than among any other association (Table 2), using the measurements to nasion would be appropriate, particularly that clinically midface position is assessed to the facial outline.

The findings reflected the potential to measure midfacial traits because of their location, and the possibility of being concordant or discrepant in determining the site of flatness or fullness. However, research is needed to further explore the correspondence between the measurements related to the maxilla and malo-zygomatic complex and panel judgments of midfacial fullness or flatness based on soft tissue profile assessments, notwithstanding the fact that the thickness of the soft tissues may minimize or exaggerate the underlying skeletal relationship. *William Arnett & Bergman (1993)* have established parameters for soft tissue midface analysis based on visual location of landmarks relative to each other, such as the cheekbone point located on a profile view at a distance of 20 to 25 mm inferior and 5 to 10 mm anterior to the outer canthus of the eye. Seemingly, the authors represent a bony structure by its soft tissue correspondence within a range of 5 mm from a landmark located at another anteroposterior plane.

The selection of landmarks to define midface position was based on the components of the midface: the naso-maxillary complex and the malo-zygomatic complex. Point J, located in the medial-central maxilla, also reflects the position of the adjacent nose, the central feature of the face. Point G represents the more lateral malo-zygomatic unit. Both landmarks are virtual *geometric* points that represent the center of the anterior maxilla (J), and the centroid of the malozygomatic triangle (G). The potential for a "center point" to be used as a proxy to identify the location of the maxilla and cheekbone is common in cephalometrics, including the location of sella turcica, symphyseal point D (*Steiner, 1953*), Xi point (*Ricketts, 1960*; *Ricketts, 1961*), and the center of the orbit (*Huertas & Ghafari, 2001*). However, J and G are derived from readily definable anatomical structures used in various cephalometric analyses (the pterygoid plate (*Ricketts, 1961*), the orbit (*Huertas & Ghafari, 2001*), the key ridge (*Ghafari, 2006*), and point A (*Steiner, 1953*; *Downs, 1949*)) with reproducible identification, as also demonstrated with the high intraclass correlation coefficients between the repeated intra- and inter- examiner identifications.

The locations of the maxillary and malo-zygomatic structures and corresponding "centroids" are not uniplanar in the frontal geometry of the face, similar to common practices in cephalometrics, in which midline structures are evaluated in the midsagittal plane (SN, SNA, SNB, ANB) and bilateral structures are assessed in other planes (Frankfort

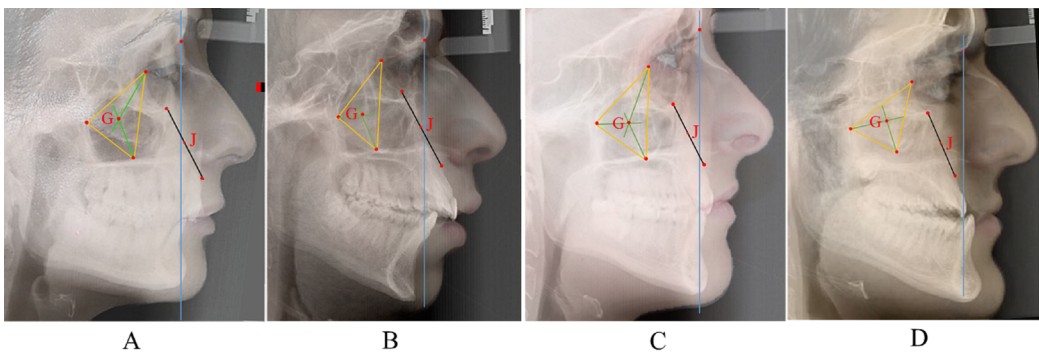

**Figure 2 Superimposed cephalometric and soft tissue facial profiles of adult patients.** Superimposed cephalometric and soft tissue facial profiles of adult patients exhibiting different midface positions based on measurements to geometric landmarks J and G, and reflecting the four different groups defined in the study. The ratios of NJ/SN and NG/SN are indicated with the corresponding interpretation: (A) NJ/SN, −7.7; Group*, <0.73; Maxillary midface, full; NG/SN, 34.4; Group*, <40.23; Zygomatic midface, full. (B) NJ/SN, 1.97; Group*, within range 0.73–6.41; Maxillary midface, normal toward full; NG/SN, 43.7; Group*, within range 40.23–44.41; Zygomatic midface, normal toward full. (C) NJ/SN, 7.17; Group*, within range 6.41–12.09; Maxillary midface, normal toward flat; NG/SN, 43.8; Group*, within range 40.23–44.41; Zygomatic midface, normal toward full. (D) NJ/SN, 18.5; Group*, >12.09; Maxillary midface, flat; NG/SN, 53.01; Group*, >48.59; Zygomatic midface, flat. * as indicated in Table 1.

horizontal, Co-Go, etc,). In all such instances, the cephalometric measurement is referenced to anatomy, itself necessarily multiplanar. The "multiplane analysis" was also described to define various levels of the multilayered face on the 2D posteroanterior cephalograph (*Ghafari, 2006*; *Grayson, McCarthy & Bookstein, 1983*). Underscoring the significance of multiplanar analyses, J and G also provided the definition of multisagittal planes.

Testing on 3D CBCT images should facilitate the study of midface position and provide more accurate measurements than the present 2D assessments. The use of CBCT imaging has not become and is not recommended as the routine cephalometric record in daily orthodontic practice because of radiation doses (*American Academy of Oral and Maxillofacial Radiology, 2013*). A recent study employing 3D CBCT images included both the maxillary and malar areas in the assessment of midface deficiency through 3D CBCT images (*Kim et al., 2018*). Although the approach was based on the definition of intersecting horizontal and vertical planes through specific landmarks on the orbit and the zygomaticomaxillary suture, this approach did not yield guidelines for routine application in individual patients. The exploration of our method in 3D CBCT images should determine this possibility. Furthermore, the application of shape analysis employed in cephalometric imaging (*Bookstein, 2016*) (e.g., tensor biometrics, finite element analysis, Procrustes method) and that could forego limitations of conventional cephalometric measurements might yield proportional assessments of the midface in its location within the face. Nevertheless, it is remarkable that differences presumably in frontal planes could be depicted on a 2D sagittal cephalograph, allowing the differentiation, at levels other than the midsagittal plane, between maxillary and malar midface position. Although the findings suggested high correlations between midface location at the maxilla and the malar bone, variations between these areas, along with the implications of multiplanar midface

measurements on facial esthetics, treatment of malocclusion, and outcome of maxillary orthognathic surgery should be investigated.

In addition, longitudinal assessment of midface position throughout growth is needed, particularly in the context of increased or decreased flatness relative to mandibular growth. Indeed, the ANB angle decreases with growth (*Van Diepenbeek, Buschang & Prahl-Andersen, 2009*) and it is possible that midface flatness might increase with such development, notwithstanding its probable increase in Class III malocclusions. Also, given the importance of proportional assessment of facial structures relative to each other, whereby absolute measurements are interpreted in the context of those of adjacent structures (*Ghafari, 2006*), future studies should be focused on the proportionality of the midface maxillary and malar positions relative to forehead and mandible. Gender differences should also be further delineated.

## CONCLUSIONS

1. The findings indicated that skeletal location of the naso-maxillary and malo-zygomatic structures can be assessed differentially. In most individuals, the locations are diagnostically similar.

2. Because the distances between either points (J or G) and other reference landmarks (nasion and sella) differ without reflecting their proportionality to head size and gender variations, the distances involving these landmarks were scaled to the cranial base length (SN) for statistical computation.

3. Measurements from nasion and sella to the maxillary and malar landmarks yielded similar findings. Accordingly, the anterior distances to nasion were adopted, reflecting the routine clinical assessment relative to the facial profile.

4. Stratification of midface position in gradation through four groups from *flat* to *full* allowed for corresponding depiction of facial measurements. Flat midfaces were characterized by canting of the cranial base and palatal plane, hyperdivergent pattern and a backward position of the maxilla (maxillary hypoplasia). The opposite was true for fuller midfaces.

5. Research is warranted in to explore midface location in relation to different variables (types of malocclusion, growing individuals, other facial structures such as forehead and mandible). Also, 3D imaging should help evaluate relationships between hard and soft tissue for proper differential diagnosis, along with the associations with facial esthetics and treatment of malocclusion.

### Funding
The authors received no funding for this work.

### Competing Interests
The authors declare that they have no competing interests.

## Author Contributions

- Chimène Chalala conceived and designed the experiments, performed the experiments, analyzed the data, contributed reagents/materials/analysis tools, prepared figures and/or tables, authored or reviewed drafts of the paper, approved the final draft.
- Joseph G. Ghafari conceived and designed the experiments, analyzed the data, contributed reagents/materials/analysis tools, prepared figures and/or tables, authored or reviewed drafts of the paper, approved the final draft.

## Ethics

The following information was supplied relating to ethical approvals (i.e., approving body and any reference numbers):

The institutional review board of the American University of Beirut gave its approval to conduct this study (IRB ID: BIO-2018-0065).

## Data Availability

The raw measurements are available as a Supplemental File.

## Supplemental Information

Supplemental information for this article can be found online at http://dx.doi.org/10.7717/peerj.8200#supplemental-information.

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
