# Peer review of "Differentiation between maxillary and malar midface position within the facial profile"

_PeerJ, doi:10.7717/peerj.8200_

## Round 0.1 · original submission · Major Revisions

My decision is major revisions, pending a satisfactory response to the issues raised

Reviewer 1 ·

Basic reporting

This manuscript is written in a clear and unambiguous style. It is well structured, with clear objectives. The literature review and the references are sufficient and pertinent to the subject of research. The conclusions are drawn from the data.

Experimental design

Appropriate

Validity of the findings

Novel and valid

Additional comments

Theodore Eliades
Academic Editor, PeerJ

Dear Dr. Eliades:
Thank you for considering me to review the article entitled “Differentiation between maxillary and malar midface position within the facial profile"
General comments:
In general, this is an interesting, well conducted and written study. It describes a novel approach (through new cephalometric landmarks and measurements) to evaluate/differentiate between fullness and flatness. In other terms, it is a 2D cephalometric study that has a 3D perspective on differentiating between the malar and the maxillary position. The study design, sample, and the performed statistical analyses were conducted in an appropriate way.
I would like the authors to consider the following comments and requests to buttress their conclusions.
1- The authors are basically evaluating flatness/fullness in adults. We do not know why they excluded growing individuals. Are there any evidence on midface flatness changing with growth? I would like the authors to add a sentence on effect of growth on midface flatness if there is any.
2- In table 1, it would be interesting to report the matching percentage among the different groups for NJ/SN and NG/SN. As an example: flat group: 90% match (only 3 individuals did not match); Normal toward flat group: 100% match..etc. And how the patients that did not match were grouped? Where did the 3 individuals in the NJ/SN flat group (that did not match with the NG/SN group) fit…etc?
Authors can report this, either in the text or in table 1 by adding one row below.
3- An interesting finding that the authors may highlight is the no difference in the maxillary length ANS-PNS between the different groups, whether categorized on NJ/SN or NG/SN. This finding implicates that the length/size of the maxilla does not play a role in the presence or not of flatness. The evaluation of flatness/fullness in this study is based on landmarks and measurement that they are higher than the palatal plane. Tis may explain why a small maxilla (in length) does not associate with flatness and a bigger maxilla with fullness.
4- Did the authors check gender differences among the different parameters? And if not, do they assume that results will not be different?
5- I suggest adding points J and G in Figures 1b-c and Figures 2A-B-C-D for better visualization.

Thank you and best regards,

Reviewer 2 ·

Basic reporting

I appreciate the courage shown by the author in conducting this research, including the time and effort they have put in. However there are certain shortcomings in this paper.

Experimental design

The authors evaluate the relative position of two geometric landmarks in the anteroposterior plane, regarding the length of the anterior cranial base and the points S and N. However midfacial flatness or fullness cannot be absolutely evaluated, i.e. without considering the position of the mandible and the forehead. The use of G point (“a more lateral landmark” according to authors) to evaluate the anteroposterior position of malar area is problematic.

Patients with craniofacial anomalies were excluded: this needs further clarification. How were the asymmetries in the Orbitale position or the zygomatic regions excluded?

Validity of the findings

Several contemporary 2- or 3D techniques are available for craniofacial shape analysis, in order to overcome the limitations of the conventional cephalometric analysis.

---

## Round 0.2 · accepted · Accept

Your revision addressed the issues raised